# Two-particle time-domain interferometry in the fractional quantum Hall effect regime

I. Taktak[1], M. Kapfer[1], J. Nath [ID][1], P. Roulleau [ID][1], M. Acciai [ID][2], J. Splettstoesser[2], I. Farrer [ID][3], D. A. Ritchie [ID][4] & D. C. Glattli [ID][1] ✉

Quasi-particles are elementary excitations of condensed matter quantum phases. Demonstrating that they keep quantum coherence while propagating is a fundamental issue for their manipulation for quantum information tasks. Here, we consider anyons, the fractionally charged quasi-particles of the Fractional Quantum Hall Effect occurring in two-dimensional electronic conductors in high magnetic fields. They obey anyonic statistics, intermediate between fermionic and bosonic. Surprisingly, anyons show large quantum coherence when transmitted through the localized states of electronic Fabry-Pérot interferometers, but almost no quantum interference when transmitted via the propagating states of Mach-Zehnder interferometers. Here, using a novel interferometric approach, we demonstrate that anyons do keep quantum coherence while propagating. Performing two-particle time-domain interference measurements sensitive to the two-particle Hanbury Brown Twiss phase, we find 53 and 60% visibilities for anyons with charges e/5 and e/3. Our results give a positive message for the challenge of performing controlled quantum coherent braiding of anyons.

The Quantum Hall Effect appears in a high perpendicular magnetic field for electrons confined to a plane. The quantization of cyclotron orbits leads to the formation of Landau levels. For integer or fractional filling of the Landau levels, a topological insulator with a gap forms. Chiral gapless modes appear on the conductor edges on which the carriers propagate, allowing a current to flow. For the integer filling factor, the quantum coherence of edge channels is large. For the fractional filling, the carriers are anyons whose quantum coherence is puzzling[1]: good coherence is observed on Fabry–Pérot interferometers[2–6] while non-existent[7–9] or weak[10,11] interference visibility is found in Mach-Zender interferometers, see the review[12]. In this work, we use a novel kind of interferometry based on two-quasi-particle Hanbury Brown Twiss (HBT) interference, which shows high quantum coherence of propagating anyons.

Fabry–Pérot interferometers (FPI), based on quantum dots or antidots, showed quasi-particle interference in the fractional quantum Hall effect (FQHE) regime as early as the 1990s through the periodic oscillations of the transmitted current versus magnetic flux or gate voltage[3]. Recently, discrete phase shifts of the interference pattern of an FPI have been reliably ascribed to the statistical angle of anyons trapped in the dot, providing, together with an independent noise experiment[13], definitive proof of anyonic statistics[2]. In an electronic Fabry–Pérot interferometer, two separate quantum point contacts (QPCs) form beam-splitters which connect a quantum dot (QD) to the right and left leads. By appropriate tuning of their transmission, the paths of carriers going straight through the two QPCs, or performing several turns inside the dot, interfere before exiting. The interference leads to the periodic oscillation of the transmission versus the magnetic flux or versus the gate voltage used to change the dot size, attesting to the quantum coherence. The resonant character of the transmission yields quasi-particle states localized in the dot with quasi-discrete spectrum. The separation between energy levels is believed to help preserve the quantum coherence needed to observe interference.

[1]Université Paris-Saclay, CEA, CNRS, SPEC, 91191 Gif-sur-Yvette, Cedex, France. [2]Department of Microtechnology and Nanoscience - MC2, Chalmers University of Technology, S-412 96 Göteborg, Sweden. [3]Department of Electronic and Electrical Engineering, University of Sheffield, Mappin Street, S1 3JD Sheffield, UK. [4]Cavendish Laboratory, University of Cambridge, J.J. Thomson Avenue, Cambridge CB3 OHE, UK. ✉e-mail: christian.glattli@cea.fr

The Mach–Zehnder interferometer (MZI) is a different kind of interferometer, also made using two beam-splitters. Its realization in electronic systems in the integer quantum Hall effect (IQHE) regime has been a breakthrough as the chiral edge channel propagation imposes a topology requiring delicate fabrication[14]. In contrast to FPIs, only two distinct paths interfere in an MZI. MZIs have been used to quantify the quantum coherence of carriers propagating along edge channels of the IQHE, in particular the loss of coherence due to the noisy environment[14,15]. Surprisingly and contrasting with measurements using FPIs, a full disappearance of interference was observed[7–9] when entering the FQHE regime (filling factor ½ < ν < 1), and only very weak interference was observed on ultra-short MZI for filling 1/3[11,12]. One possible reason for the different behavior could be ascribed to the nature of states involved in the two interferometers: discrete versus continuous spectrum, the latter being more fragile with respect to environmental fluctuations[16]. Counter-propagating neutral modes are also believed to degrade the coherence. Fundamental anyonic phase fluctuations may also impact the MZI visibility. The puzzling MZI visibility requires searching for quantum interferences using a different kind of interferometer, also based on delocalized propagating states.

This is the aim of this work. Here, we use a single beam-splitter to perform time-controlled quantum interference of propagating quasiparticles in the IQHE and FQHE regime. While FPI and MZI interferometers, based on DC conductance measurements, are sensitive to single-quasi-particle interference, our measurements detect current fluctuations (quantum shot noise), which is known to reveal two-particle interference, see refs. [17–19]. The experiment is inspired by the seminal work of ref. [20] who proposed a new kind of interferometry, which is sensitive to the measurement of the so-called Hanbury Brown Twiss (HBT) quantum phase. They showed that, when applying two phase-shifted ac potentials of equal magnitude and frequency on two different contacts of a four-probe conductor, the current noise reveals a two-particle interference resulting from particle indistinguishability. Here, we use the simplest four-terminal conductor: a QPC, which mixes and partitions two incoming chiral edge channels into two transmitted and reflected edge channels. Figure 1a shows the principle of the two-particle time-domain interferometry measurement and Fig. 1b is a sketch of the experimental set-up used. The experiments are first done in the IQHE regime at filling factor ν = 2, as a benchmark situation, and then in the FQHE regime at filling factor ν = 2/5, allowing us to probe anyons with charges e* = e/5 and e/3. Note that, the ν = 2/5 FQHE state

maps to the ν = 2 IQHE state in the composite fermion picture[21] as both filling factors have two co-propagating edge channels.

Two AC sinewave voltages $V(t) = V_{ac}\cos(2\pi t)$ and $V(t - \tau) = V_{ac}\cos(2\pi t - \Delta\Phi)$ are applied to contacts (1) and (2) respectively to inject photo-created electron-hole pairs in the beam-splitter input leads, see Fig. 1a. $\Delta\Phi = 2\pi f\tau$ is the relative phase-shift due to the time delay τ between the sources. The scattering amplitudes relating input leads (1) and (2) to output leads (3) and (4) are $s_{3,1} = s_{4,2} = t$ and $s_{3,2} = s_{4,1} = ir$ (to make expressions simpler, in the main text, we disregard in the scattering amplitudes the phase factors $e^{i\varphi_{\beta\alpha}}$, where $\varphi_{\beta\alpha}$ is the phase accumulated by electrons propagating from input contact (α) to output contacts (β)). $|t|^2 = D$ and $|r|^2 = 1 - D$ are the transmission and reflection probabilities of the QPC beam-splitter. Electron and hole interferences are detected through the cross-correlated current fluctuations of leads (3) and (4). According to ref. [20], the magnitude of the current cross-correlation is shown to depend on the Hanbury Brown Twiss phase $\chi = \arg(s_{13}^* s_{32} s_{24}^* s_{41})$ resulting from the indistinguishability of photo-created electron-hole pairs. Contrasting with pure DC transport and noise experiments, the creation of photo-assisted electron-hole pairs allows us to probe the HBT phase, providing a novel interferometry to test the quantum coherence of carriers. This is done by varying the time delay between the AC sources and measuring the cross-correlation noise. Note that the existence of the HBT phase is only important to probe the coherence. Its absolute value, however is not relevant, being sample dependent like the geometrical phase of an MZI. The cross-correlation noise expression, in the limit of single-photon excitation, is, for $eV_{ac} \ll hf$,[20]:

$$S_{I_3 I_4} = -e^2 f\left(\frac{eV_{ac}}{2hf}\right)^2 \left| s_{13}^* s_{41} e^{-i\Delta\phi} + s_{23}^* s_{42} \right|^2 \qquad (1)$$

Expression (1) contains the square of the sum of two two-particle probability amplitudes corresponding to the two possible electron-hole paths (a) and (b) shown in Fig. 1a, whose indistinguishability is controlled by the time delay τ. The process where an electron arrives in lead (4) and a hole in lead (3) gives similar interference and these two processes lead to current fluctuations, hence the noise given by (1).

The time delay between AC voltages provides the knob to modulate the interference between the two paths. This plays the role of the magnetic flux or the gate voltage control used to vary the interference in FP and MZ interferometers. The complete expression, based

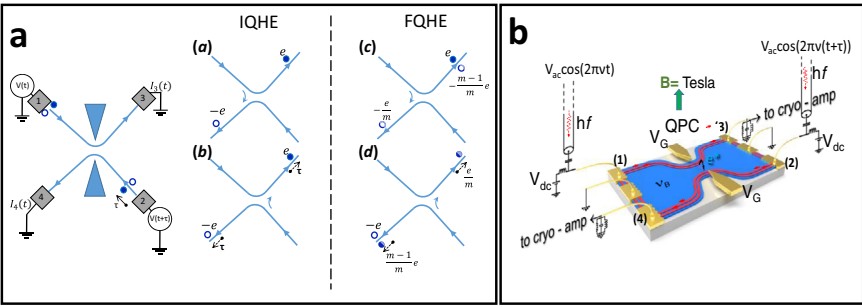

**Fig. 1 | Two-particle dynamical interference principle and schematic experimental implementation. a** Electron-holes pairs emitted by the AC-biased contacts (1) and (2) are scattered into contacts (3) and (4) by a tunable QPC which forms an electronic beam-splitter. In the IQHE regime, there are two scattering processes, where, in (a), the electron-hole pair is created in lead (1) and where, in (b), the electron-hole pair is created in lead (2). They both lead to an electron in (3) and a hole in (4). Indistinguishability leads to a probability of occurrence of processes (a) and (b) whose variation with the relative time delay τ reveals two-particle interference. Two similar interfering scattering processes (not shown) lead to a hole in (3) and an electron in (4). This generates the cross-correlated current noise $S_{I_3 I_4}$ which is measured in the set-up sketched in Fig. 1b. In the FQHE regime, the

scattering processes are similar but involve e/m fractionally charged anyons (m = 3 or 5). Photo-created electron-hole pairs give rise to the scattering of two indistinguishable processes where in (c) ((d)), an electron (hole) is transmitted while a hole (electron) is backscattered as a charge −e/m (e/m) and transmitted as charge −(m − 1)e/m ((m − 1)e/m), respectively. **b** Sketch of the measurement principle. Two separate coaxial lines bring the microwave excitation on contact (1) and (2) to generate electron-hole pairs. The fluctuations of the transmitted and reflected current are converted into voltage fluctuations at contact (3) and (4). After frequency filtering and cryogenic amplification, the cross-correlated noise spectrum is computed.

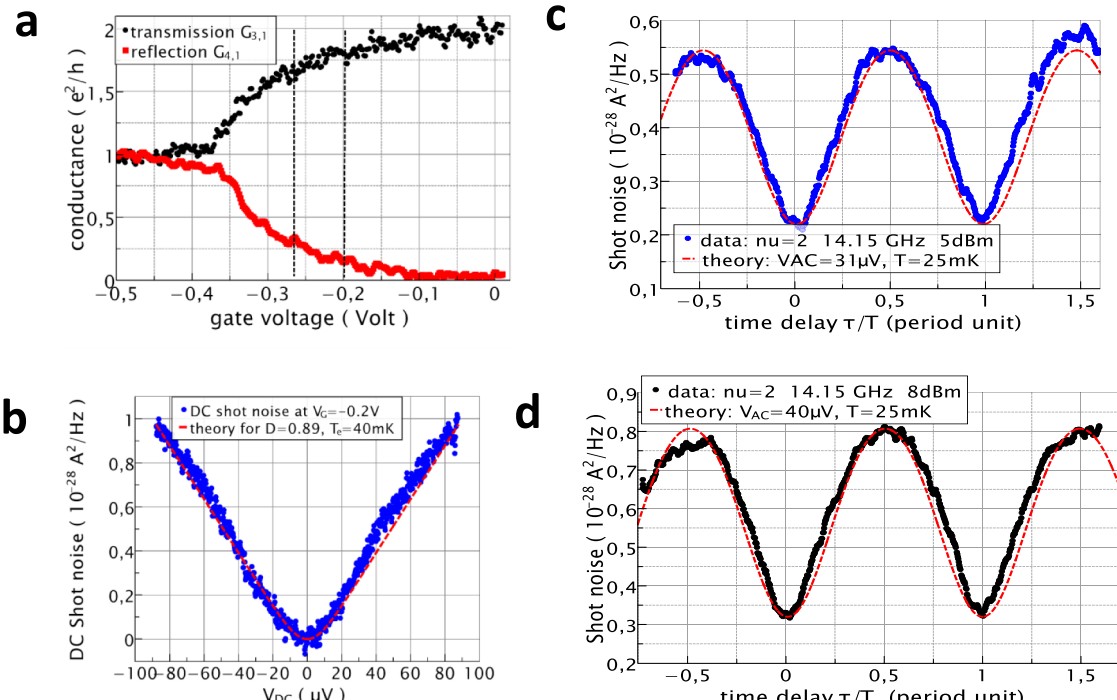

**Fig. 2 | Two-particle interferometry in the IQHE regime. a** DC conductance trace versus gate voltage is used to measure the transmission and reflection of the inner edge channel while the outer edge channel is transmitted at filling factor $v = 2$. The vertical dashed lines denote the working points used here. **b** DC Cross-correlated noise ($-S_{I_3 I_4}$) data (blue circles) versus DC voltage bias $V_{DC}$ and comparison with Eq. (4) for transmission $D = 0.89$ and temperature 40 mK (red dashed line). The expected statistical error is $\pm 4.2 \times 10^{-30}$ A²/Hz for 1.5 s measurement time per DC voltage bias point. **c, d** Blue and black circles data points show the shot noise measured versus the time delay for 14.15 GHz microwave excitation and RF source powers 5 and 8 dB, respectively. The dashed red curves provide a comparison with the heuristic model given by Eq. (3), including the noise offset $S_M$. The statistical noise errors error bars are expected $\pm 3.10^{-30}$ A²/Hz for 3 s measurement time.

on photo-assisted shot noise and including finite temperature and multiphoton absorption/emission of electrons and holes, is:

$$S_{I_3 I_4} = -2e^2 f D(1-D) \sum_{l=+\infty}^{-\infty} l \left[ J_l \left( \frac{e V_{ac}}{hf} 2 \sin\left(\frac{\Delta \Phi'}{2}\right) \right) \right]^2 \\ \times \left( \coth\left(\frac{lhf}{2k_B T}\right) - 2k_B T / lhf \right),$$ (2)

where $\Delta \Phi' = \Delta \Phi \cdot \chi + \pi$. For small $V_{ac}$ and zero temperatures, this expression is equivalent to Eq. (1).

## Results

### IQHE regime

The measurements are performed on a two-dimensional electron gas made in high mobility GaAs/Ga(Al)As heterojunction with electron density $n_s = 1.11 \times 10^{15}$ m$^{-2}$ and zero field mobility $\mu = 300$ Vm² s$^{-1}$. The filling factor $v = 2$ occurs at field B = 2.3 T. Low-frequency conductance measurements, done while varying the QPC gate voltage $V_G$, are first performed to measure the transmitted and reflected differential conductance by applying a small (few μV) amplitude 270 Hz frequency AC voltage on contact (1). The conductance traces are shown in Fig. 2a. The $e^2/h$ conductance plateau for $V_G < -0.37$ volts signals the full reflection of the $v = 2$ inner edge channel. We choose to partition the inner edge channel only while the outer edge channel is fully transmitted. Similar observations can potentially be obtained when fully reflecting the inner edge and partitioning the outer edge. We, therefore, concentrate on two working points at $V_G = -0.2$ and $-0.27$ V, respectively, corresponding to partial transmissions $D = 0.89$ and 0.84 of the inner edge channel.

Figure 2c, d show two-particle interference noise measurements for a microwave frequency of 14.15 GHz at $V_G = -0.2$ V for two microwave excitations differing in power by 3 dB power of the sources. For the chosen frequency and 20 mK temperature, the photon energy is $hf \approx 35.k_B T$ so that thermal effects are small. Clear two-particle interference is observed when varying the time delay. The visibility, calculated by the ratio of the difference between the maximum and the minimum noise over the sum of the maximum and minimum, is found to be 43 and 44% for both microwave excitations. Measurements at smaller QPC transmission, $V_G = -0.27$ V, for two different AC excitations also differing in power by 3 dB give similar 38 and 40% visibility, see Supplementary Note A4.

The above measurements demonstrate two-particle quantum interference in the IQHE regime. The electron-hole pair interference can be put in perspective with electronic Hong Ou Mandel experiments where, instead of electron-hole pairs, single electrons are periodically emitted by an on-demand single electron source[22–25]. In ref. [24], a driven quantum dot in the IQHE regime (also $v = 2$) injecting single electrons in the outer edge of a QPC beam-splitter gave the first evidence of two-particle dynamical interference. Later, Hong Ou Mandel interference with 100% visibility was observed at zero magnetic field in ref. [25] using a different on-demand single electron source generating single particle states called levitons. In ref. [24] electrons were injected on the outer edge only and the low visibility was attributed to the Coulomb interaction between inner and outer co-propagating edge channels giving spin-charge separation. In the present experiment, a 100% visibility may have been expected from Eq. (2). Indeed, as theoretically shown in refs. [26–29], no loss of visibility is expected when including Coulomb interaction between co-propagating edge channels as the AC voltages generate coherent states which do not suffer from decoherence due to inter-channel Coulomb interaction. To understand the finite but reduced 40% visibility observed here, we suggest that this may be due to the weak elastic mixing of co-propagating channels[29] resulting from local impurities combined with spin-orbit

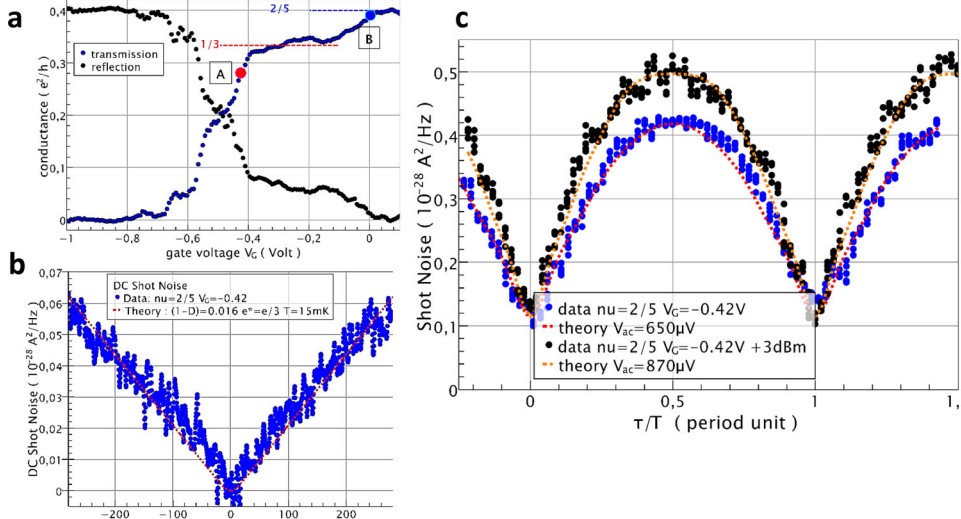

**Fig. 3 | Two-particle dynamical interference of e/3 anyons in the FQHE regime.**
**a** QPC conductance data versus gate voltage $V_G$ at field B = 11.3 Tesla ($\nu$ = 2/5). A plateau at conductance $\frac{e^2}{3h}$ signals the reflection of the first inner 2/5 edge channel and the formation of a 1/3 fractional state in the QPC. The red and blue filled circles, labeled A and B, indicate the working points used for probing the coherence of respectively e/3 and e/5 anyons. **b** DC cross-correlated noise (($-S_{I_3 I_4}$) data (blue circles), measured at working point A, and comparison with Eq. (7) for constant transmission D = 0.984, temperature 15 mK and e* = e/3 (red dashed curve). The

statistical measurement noise errors are $\pm 0.8 \times 10^{-30}$ A$^2$/Hz for the typical 15 s measurement time. **c** Shot noise data (black and blue filled circles) versus time delay measured at working point A for a 14.15 GHz microwave excitation and two RF source powers differing by 3 dB. The red dashed lines are comparisons with Eq. (6) using $e^* = \frac{e}{3}$ including a small $10^{-29}$ A$^2$/Hz noise offset. The statistical measurement noise errors are $\pm 2 \times 10^{-30}$ A$^2$/Hz for the 3 s measurement time per time delay point used.

coupling. Indeed, for different inner and outer edge channel velocities, the HBT phase for electrons injected in the inner edge and reaching the QPC in the outer edge differs from the HBT phase of electrons emitted in the outer edge channel and remaining in this channel. One can also say that the tunneling between the outer and inner edge input leads allows the inner edge channel to gain information on "which-path" the incoming electron-hole pairs are taking, thus breaking indistinguishability. In Supplementary Note A4 we give reasonable numbers supporting this, based on the channel mixing modeling of ref. 29 briefly recalled in Supplementary Note B2. The loss of coherence can be brought in correspondence with the one observed in "which-path" experiments done with a standard MZ interferometer[30]. Further studies addressing a systematic analysis of visibility versus edge channel length should confirm the mixing hypothesis that we put forward here. We leave this issue for future works.

In order to perform a quantitative analysis, one can remark that Eq. (2) can be expressed as a sum of DC shot noises, i.e., shot noise without microwave excitation, where $V_{DC}$ is replaced in the expression by $lhf/e$ and weighted by the term $[J_l(\frac{eV_{ac}}{hf}2\sin(\frac{\Delta\Phi}{2}))]^2$

$$S_{I_3 I_4} = S_M + \sum_{l=+\infty}^{-\infty} \left[ J_l \left( \frac{eV_{ac}}{hf} 2\sin\left(\frac{\Delta\Phi'}{2}\right) \right) \right]^2 S_{I_3 I_4}^{DC}\left( V_{DC} = \frac{lhf}{e}, T \right) \quad (3)$$

with:

$$S_{I_3 I_4}^{DC}(V_{DC}, T) = -2e^2 D(1-D)eV_{DC}\left( \coth\left(\frac{eV_{DC}}{2k_B T}\right) - 2k_B T/eV_{DC} \right) \quad (4)$$

In Eq. (3) we have added an extra noise term $S_M$ which may account for a possible mixing of co-propagating edge channels, see below. This heuristic approach provides the best fit for data. As explained in the Supplementary Note B2, it is supported by channel mixing hypothesis[29] which leads to a closed expression, see Supplementary Eq. (S6), where a mixing noise contribution is found not to depend on $\tau$, like the heuristic term $S_M$ in Eq. (3). Elastic tunneling between co-propagating edge channels is likely to occur. We found a

mixing tunneling probability of about 10% for our 18 µm incoming channel length. This indicates a few 100 µm scattering lengths which is compatible with typical scattering lengths ranging from a few tens of µm to 100 µm reported at filling factor 2[31–33].

To analyze the noise interference data, we use the DC shot noise measurements versus $V_{DC}$ applied on contact (1) for the gate voltage $V_G = -0.2$ V shown in Fig. 2b. The red dashed line compares the data to Eq. (4) using the known transmission. Knowing the DC shot noise parameters extracted from the dashed red curve fit of Fig. 2b, microwave frequency and phase difference, we are left with only two unknown parameters, $V_{ac}$ and $S_M$, to analyze and fit the two-particle noise interference using the heuristic model (3).

The red dashed curves in Fig. 2c, d are best fits using Eq. (3). One finds $V_{ac} = 31$ µV, $S_M = 2.1 \times 10^{-29}$ A$^2$/Hz, and 40 µV, $S_M = 3.2 \times 10^{-29}$ A$^2$/Hz, respectively for the two different excitations differing by a 3 dB power increment. Similar fits for $V_G = -0.27$ V give $V_{ac} = 33.5$ and 41 µV, respectively, using the heuristic formula (3). Slightly higher amplitudes are found using the complete channel mixing model of ref. 29 (see Supplementary Note A4). For both gate voltages, the ratio of the microwave amplitudes for 3 dB power increment is close to √2, albeit slightly lower, giving confidence in the analysis. We now turn to the investigation of two-particle dynamical interference in the FQHE regime.

## Results in the FQHE regime

We concentrate on the so-called weak backscattering regime $1 - D \ll 1$ for which the quasi-particle scattering is best understood. Consider, for simplicity, an edge channel for which the tunneling excitations carry a charge e/3. Similar reasoning can be done for charge e/5. Electrons in the edge channel form a correlated one-dimensional quantum liquid, densely occupying one state over three. To the lowest order in the backscattering amplitude, an electron can be either transmitted as a whole, charge e, with amplitude of probability $t \approx 1$, or be split as a backscattered charge e/3 and a transmitted charge 2e/3 with amplitude of probability $ir$. The same can occur for a hole, with respective charges −e, −e/3, and −2e/3 and amplitude $(ir)^*$. Now consider an electron-hole

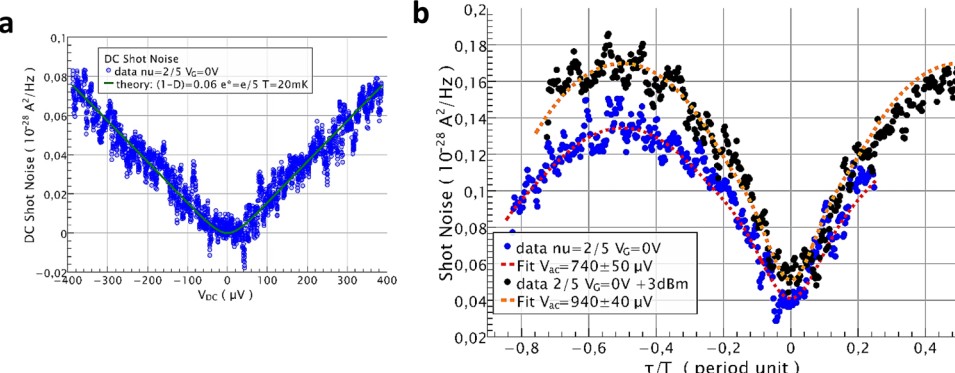

**Fig. 4 | Two-particle dynamical interference of e/5 anyons in the FQHE regime.**
**a** DC cross-correlated noise ($-S_{I_3 I_4}$) data (blue circles), recorded at working point B. The red dashed curve is computed from Eq. (7) for transmission $D = 0.94$ and $T_{el.} = 20\,mK$ (green solid curve) and e* = e/5. Statistical measurement noise errors are expected to be $\pm 11 \times 10^{-30}\,A^2/Hz$ for the 10 s measurement time. **b** The black and blue filled circles are shot noise measurement data versus time delay taken at working point B for 14.15 GHz microwave excitation and two RF source powers differing by 3 dB. The red dashed curves are computed from Eq. (6) using e* = e/5 and temperature 20 mK, including a $4.10^{-30}\,A^2/Hz$ noise offset. The statistical measurement noise errors are $\pm 0.9 \times 10^{-30}\,A^2/Hz$ for the 15 s acquisition time per time delay point.

pair created in input lead (1) and focus on the event where the hole is backscattered. After scattering, we are left with a charge $e$ transmitted to output lead (3) and a split hole transmitted as charge $-2e/3$ in lead (3) and reflected as hole of charge $-e/3$ in lead (4), see Fig. 1c. This process leads to a total charge $e/3$ in lead (3) and $-e/3$ in lead (4). The same set of charges in the output contacts is also obtained for the process where an electron-hole pair is created in input lead (2) and the electron is backscattered. After scattering, the electron is split as a charge $e/3$ in lead (3) and transmitted as a charge $2e/3$ in lead (4) while the hole is transmitted as charge $-e$ in lead (4) as shown in Fig. 1d. When only charge is considered, the two events of Fig. 1c, d are not distinguishable and interfere.

Another process, leading to charge $-e/3$ and $e/3$ in the output leads (3) and (4), respectively, give similar interference. The anti-correlated fluctuations of $\pm e/3$ charges give rise to a noise whose expression is similar to Eq. (1). When written to first order in reflection probability 1-D, it yields, disregarding the propagation phase accumulated in the leads:

$$S_{I_3 I_4} = -4(e^*)^2 f(1-D)\left(\frac{e^* V_{ac}}{2hf}\right)^2 |1 - e^{-i\Delta\Phi}|^2 \quad (5)$$

This expression agrees with the low $V_{ac}$ limit of a full multiphoton expression which has been theoretically obtained using perturbative approaches, including interactions[34–36]:

$$S_{I_3 I_4} = \sum_{l=+\infty}^{-\infty}\left[J_l\left(\frac{e^* V_{ac}}{hf} 2\sin\left(\frac{\Delta\Phi'}{2}\right)\right)\right]^2 S_{I_3 I_4}^{DC}\left(V_{DC} = \frac{lhf}{e^*}, T\right) \quad (6)$$

where:

$$S_{I_3 I_4}^{DC}(V_{DC}, T) \approx -2(e^*)^2(1-D)e^* V_{DC}\left(\coth\left(\frac{e^* V_{DC}}{2k_B T}\right) - 2k_B T / e^* V_{DC}\right) \quad (7)$$

The expressions are similar to those of the integer regime Eq. (4), except for the quasi-particle charge e* and the limit 1-D«1.

Figure 3a shows the conductance of the QPC for filling factor $\nu = 2/5$ in bulk ($B = 11.3\,T$). Starting at $\frac{2}{5}e^2/h$ for $V_G = 0.2\,V$, the conductance decreases to reach a plateau at $\frac{1}{3}e^2/h$ for $V_G < -0.1\,V$ signaling the reflection of the 2/5 inner edge channel and the formation of a region of filling factor $\nu_G = 1/3$ inside the QPC. For $V_G < -0.38\,V$, the last edge channel starts to be reflected. We choose two working points A and B, at $V_G = -0.42$ and $-0.0\,V$, respectively, corresponding to the weak

backscattering of anyons with charge e* = e/3 and e* = e/5, respectively. These fractional charges have been previously confirmed in a similar regime through the measurements of their Josephson relation in previous work, see ref. 37. For the weak backscattering regime of the $\nu_G = 1/3$ state, at working point A, Fig. 3c shows the shot noise data versus time delay for microwave powers differing by 3 dB and microwave frequency f = 14.15 GHz. Clear oscillations with 55 and 60% visibility are observed.

To analyze the data, we introduce a constant noise offset $S_M$ to take into account a possible mixing noise, as done for the IQHE study. The fits, calculated from Eq. (6) with e* = e/3 and using the measured DC shot noise data taken in the same conditions, are shown as red dashed curves. They provide an excellent agreement with each experimental trace with $V_{ac} = 650 \pm 20\,\mu V$ and $V_{ac} = 870 \pm 10\,\mu V$ and $S_M = 0.11 \times 10^{-28}\,A^2/Hz$. For a 3 dB microwave power difference, the ratio of the $V_{ac}$ values is $1.34 \pm 0.05$, close to $\sqrt{2}$. Note that the theoretical analysis would have required DC shot noise data up to $eV_{DC} \cong 10hf$ while the range of measurements in Fig. 3b is limited to $V_{DC} \cong 7hf$. Such extrapolation has been also safely used in ref. 37 for similar conditions.

Similar measurements performed for $\nu_G = 2/5$, at working point B, are displayed in Fig. 4b for the same 14.15 GHz frequency and two microwave powers differing by 3 dB. Visibilities of 53 and 51% are observed. The fits done using e* = e/5 and experimental DC shot noise traces give $V_{ac} = 740 \pm 50\,\mu V$ and $V_{ac} = 940 \pm 40\,\mu V$, with a $V_{ac}$ ratio $1.27 \pm 0.15$ weaker but still close to $\sqrt{2}$ for a 3 dB power difference. Here the weaker noise due to the one fifth charge leads to larger uncertainty due to a lower signal-to-noise ratio.

The fact that higher visibility is found at filling factor 2/5 than at filling factor 2 may be, at first sight surprising. It indicates that less channel mixing occurs between co-propagating edge channels in the FQHE regime than in the IQHE regime. Note, however, that no direct comparison can be done between these two cases since the underlying mixing mechanisms are likely to be different. Possible reasons for this are the spin polarization at 11 T ($\nu = 2/5$), which cannot be compared with that at a field of 2.3 T ($\nu = 2$), as well as the reduced inter-channel tunneling at low energies, which is expected from chiral Luttinger liquid physics[38]. These effects motivate further theoretical modeling.

To conclude, the present demonstration of two-particle dynamical interference in the FQHE regime shows that the propagating edge channels do keep significant quantum coherence over several tens of microns (our sample dimension), contrasting with the experiments performed using MZIs. The work also provides the first example of

novel interferometry based on the HBT phase. This work helps resolve the issue of diverging results previously reported on quantum coherence in the FQHE regime, at the same time highlighting the work needed on MZI to understand the prior results better. The current demonstration also highlights the possibility of performing anyon braiding in FQHE systems, which is a crucial step towards the realization of topologically protected qubits for quantum computing.

## Methods

Experiments are done in a Cryoconcept dry dilution refrigerator with a 20 mK base temperature equipped with a 14.5 T dry superconducting magnet. Conductance measurements are done using lock-in amplifiers at 270 Hz frequency and 2 μV excitation voltage. The two microwave excitations used for interferometry are provided using two synchronized DC-40GHz room temperature sources with relative phase control ability. About 60–70 dB microwave attenuation between the source and the sample is provided by cryogenic 50 Ohms attenuators. Noise measurements are made using homemade cryogenic amplifiers followed by fast digital acquisition. The FFT cross-correlation computation is done on a PC computer providing real-time noise acquisition.

## Data availability

All data, code, and materials used in the analysis are available in some form to any researcher for purposes of reproducing or extending the analysis. The raw experimental shot noise data of all figures are available as a single zip file at https://doi.org/10.5281/zenodo.6796840.

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

## Acknowledgements

We thank P. Jacques for technical help, P. Pari, P. Forget, and M. de Combarieu for cryogenic support, and we acknowledge discussions with J. Rech, Th. Jonkheere, A. Crépieux, I. Safi, Th. Martin, Y. Gefen, M. Goldstein, A. Das, S. Manna, and members of the Saclay Nanoelectronics team. This work was funded by the ANR FullyQuantum 16-CE30-0015-01 grant, the H2020 FET-OPEN UltraFastNano #862683 grant, and the EPSRC grant K004077.

## Author contributions

D.C.G. designed and supervised the project. I.T. performed the experiment with help from D.C.G. and J.N.; M.K. fabricated the sample on heterojunctions grown by I.F. and D.A.R. I.T., D.C.G., J.S., M.A., and P.R. analyzed and discussed the data I.T. and D.C.G. wrote the manuscript with inputs from all co-authors.

## Competing interests

The authors declare no competing interests.
