## [Peer Review File · Nature Communications]

REVIEWER COMMENTS

Reviewer #1 (Remarks to the Author):

Report on the manuscript NCOMMS-22-03842-T "Two-particle time-domain interferometry in the Fractional Quantum Hall Effect regime" by I. Taktak et al.

The authors experimentally study two-particle interferometry in the time domain, both in the integer and fractional quantum Hall regime. In a beautiful experiment, they indeed find an interference signal in the fractional quantum Hall regime, which they interpret as evidence for the possibility of coherent processes. The experiment and its theoretical interpretation seem sound to me.

The topic of the experiment is of considerable current interest, given that quantum Hall interferometry has been able to provide evidence for anyonic exchange statistics and holds the promise to study possible non-abelian statistics in the $5/2$ and related states. I warmly recommend the manuscript for acceptance once the authors have addressed my comment below.

The authors say that their results are a positive indication for the feasibility of fractional Mach-Zender interference (MZI). However, the discussion of what the present experiment implies for the prospect of Mach-Zender interferometry in the fractional quantum Hall regime is quite vague. The authors say that their experiment proves that coherent processes are possible even in the fractional regime. However, even in light of the recent fractional FPI from the Weizmann Institute (arXiv:2203.04205), there is still a strong contradiction of findings: in the FPI case, only interference of an outer edge channel is experimentally feasible, while in the present experiment both inner and outer edge channels display coherence. Specifically, I would like to ask the authors to estimate the degree of time and spatial coherence which is required for their time domain experiment, and compare these estimates to the requirements for successfully observing MZI interference.

Reviewer #2 (Remarks to the Author):

The paper by Taktak et al. describes the interference of quasiparticles in the fractional quantum Hall regime. This field has recently gained renewed interest, since quasiparticle interference in a Fabry-Perot

interferometer was successfully used to directly measure the fractional statistics of these quasiparticles. For instance, a very recent review article (which is not mentioned in this version of the present paper) on this topic appeared in Nature Reviews Physics: <https://doi.org/10.1038/s42254-021-00351-0>. Thus, the paper by Taktak et al. addresses a topic of relevance for the entire condensed matter community involved in quantum Hall physics and topological states of matter. However, before I can recommend this manuscript for publication, a major revision is required, which convincingly addresses the following points.

It is rather puzzling that the visibility of the interference oscillations in the fractional quantum Hall regime (up to 60%) is larger than in the integer quantum Hall regime (about 40%). In my opinion, the authors have to convincingly explain this result, because if left unexplained, it would raise doubts about the general validity of their analysis.

Moreover, the authors do discuss the low visibility in the integer quantum Hall regime and explain it with a mixing between co-propagating channels. This could be easily checked by measuring the visibility at filling factor 1, where only a single edge channel is present. These additional measurements would provide a more robust ground for the proposed explanation. Besides, it is not explained why this process should be less relevant in the fractional QH regime, where for filling factor $2/5$ there are again two co-propagating and interacting channels. In addition, it is stated, correctly, that Coulomb interactions for filling factor 2 cannot explain a reduced visibility, which is then ascribed to some interaction with impurities and spin-orbit interactions. However, some further discussion needs to be added: the GaAs/GaAlAs 2DEG material does not have a strong spin orbit component (in contrast to InAs or InSb); the mentioned theory reference deals with a single impurity and two channels, while here one of the two channels is fully reflected; and in addition, I would expect an impurity distribution.

In the introduction and the conclusions, the authors underline that their results will provide guidance to understand previous results on interferometry in Mach-Zehnder (MZ) geometry in the fractional QH regime. However, this argument is rather weak, as in one case, two-quasiparticle interference is detected, while in the other, single-quasiparticle interference. Thus, the argument should be elaborated better. In addition, it is written that in MZ interferometers no oscillations in the fractional quantum Hall regime have been reported so far. This is somehow in contrast with the findings of Deviatov et al. (doi: 10.1209/0295-5075/100/67009) that were performed in the fractional regime: could the authors comment on this point?

More in detail:

I suppose there is some mix-up in Figure 1(f). Contact (2) should be biased but is indicated as grounded.

Page 4, line 20, the unit of mobility is m^2/Vs .

On page 4, the authors state that “We choose to only partition the inner edge channel while the outer edge channel is fully transmitted. Similar observations can be obtained when fully reflecting the inner edge and partitioning the outer edge.” These results should be shown in the SI.

On page 6, when discussing typical scattering length for elastic tunneling between co-propagating edge channels, it would help to give the device size for comparison (which is only provided in the SI).

At the beginning of the part on the FQHE regime, I wonder how useful it is to discuss the $\nu=1/3$ example when the experiments were performed at $\nu=2/5$ in the bulk. In addition, it is true that using a composite fermion picture, $\nu=2/5$ can be mapped onto $\nu=2$ filling with dressed entities. However, I think that this might be too trivial for the $\nu=2/5$ physics. For instance, the non-Fermi liquid behavior expected for fractional edge states results in a peculiar, and well-known, power-law behavior that is not mentioned at all. I think that the theory side should be thus revised, to avoid a too simple picture.

Reviewer #3 (Remarks to the Author):

This paper describes a Hanbury Brown and Twiss (HBT) experiment for fractional quasiparticles in the fractional quantum Hall regime. The cross-correlation noise under microwave excitations shows sinusoidal dependence on the time delay between the two microwave sources. The visibility is high (about 40% for integer charges and 60% for fractional charges), and this degradation is explained by “mixing” (tunneling) between the copropagating channels. The two-particle interferometry for fractional charges is demonstrated with a high-quality data set. This provides a novel characteristic of fractional charges. I have several questions before recommending this paper for publication in Nature Communications.

While the demonstrated scheme should ‘probe the HBT phase’ as shown in the text, this is not discussed with the experimental data. The authors should describe theoretically predicted values and experimentally demonstrated values of the HBT phase.

For the fractional case, the ac amplitude of 650 and 870 μV in Fig. 3(c) is much larger than the voltage range (up to 250 μV) used for the DC shot noise in Fig. 3(b). The authors should confirm that fractional

charge tunneling is allowed even for the large bias. The maximum bias can be $2\sqrt{2}$ times the rms amplitude.

The supplementary data in Fig. S2 shows that the dip position shifts with the P2 power. Why is this?

Two notations, Δ/Fai and Δ/fai , are used for the relative phase.

Reviewer #1 (Remarks to the Author):

R#1: *The authors experimentally study two-particle interferometry in the time domain, both in the integer and fractional quantum Hall regime. In a beautiful experiment, they indeed find an interference signal in the fractional quantum Hall regime, which they interpret as evidence for the possibility of coherent processes. The experiment and its theoretical interpretation seem sound to me. The topic of the experiment is of considerable current interest, given that quantum Hall interferometry has been able to provide evidence for anyonic exchange statistics and holds the promise to study possible non-abelian statistics in the 5/2 and related states. I warmly recommend the manuscript for acceptance once the authors have addressed my comment below.*

Answer 1.1: We thank Reviewer#1 for the positive comments regarding the relevance of our work addressing the important issue of quantum coherence in the FQHE regime.

R#1: *The authors say that their results are a positive indication for the feasibility of fractional Mach-Zehnder interference (MZI). However, the discussion of what the present experiment implies for the prospect of Mach-Zehnder interferometry in the fractional quantum Hall regime is quite vague. The authors say that their experiment proves that coherent processes are possible even in the fractional regime. However, even in light of the recent fractional FPI from the Weizmann Institute (arXiv:2203.04205), there is still a strong contradiction of findings: in the FPI case, only interference of an outer edge channel is experimentally feasible, while in the present experiment both inner and outer edge channels display coherence. Specifically, I would like to ask the authors to estimate the degree of time and spatial coherences which is required for their time domain experiment, and compare these estimates to the requirements for successfully observing MZI interference.*

Answer 1.2: If we correctly understand, Reviewer#1 suggests that a typical coherence time/length can be extracted from our two-particle dynamic HBT interference measurements. However, as written in the original main text, we think that “a systematic study of visibility versus edge channel length is required” to give a rigorous answer to this important issue. At the present stage, the aim of our manuscript is to inform the community that one observes a high coherence in the FQHE regime for propagating quasiparticles, which contrasts the first experiments done in MZIs.

Furthermore, we thank Reviewer#1 for mentioning reference arXiv:2203.04205 where finite coherence in FQHE is observed using a MZI of very short length. The recent Weizmann experiment is very encouraging. However, due to the topology of the chiral MZI, anyons entering the inner contact give rise to a fundamental $(0, \pi/3, -\pi/3)$ phase jittering which strongly reduces the MZI visibility, see Eq.(2) of arXiv:2203.04205 . This makes the extraction of single particle coherence length not straightforward, requiring more theoretical developments, and this makes direct comparison with simpler interferometers difficult. In our present two-particle interferometer, the topology does not lead to anyonic phase jittering and therefore the visibility is expected to be much higher.

We have included reference arXiv:2203.04205 and have weakened, in the main text the statement about the absence of interference of MZI in the FQHE regime.

We now reply to the comment of Reviewer#1 regarding a possible comparison between the actual visibility measured in our two-particle dynamical HBT interferometer and that

expected for a single-particle Mach-Zehnder interferometer. In response to this comment, we have added in the supplementary material (B3) a calculation of the expected visibility when the losses due to similar channel mixing occur. For channel mixing strength $R_A=R_B=0.1$ in each arm, the MZI visibility is found 90% for optimal MZI tuning, showing that also the MZI visibility is affected by mixing and is not necessarily only reduced by a limited phase coherence. In the present experiment, we observe instead 40%. However, we think that this comparison is not fully representative, since we compare two-particle and single-particle interference, and it is known that for MZIs another important source of decoherence is due to the thermal fluctuations of the Coulomb coupled copropagating channel, a source of decoherence not expected in our new type of interferometry (see [28]).

Reviewer #2 (Remarks to the Author):

R#2: *The paper by Taktak et al. describes the interference of quasiparticles in the fractional quantum Hall regime. This field has recently gained renewed interest, since quasiparticle interference in a Fabry-Perot interferometer was successfully used to directly measure the fractional statistics of these quasiparticles. For instance, a very recent review article (which is not mentioned in this version of the present paper) on this topic appeared in Nature Reviews Physics: <https://doi.org/10.1038/s42254-021-00351-0>. Thus, the paper by Taktak et al. addresses a topic of relevance for the entire condensed matter community involved in quantum Hall physics and topological states of matter. However, before I can recommend this manuscript for publication, a major revision is required, which convincingly addresses the following points.*

Answer 2.1: We thank Reviewer#2 for the positive comments regarding the relevance of our work and for pointing us the interesting review (*Nature Reviews Physics*, 3, 698–711 (2021)). We have included this review in the references of the revised manuscript.

In the following, we address the detailed comments by Reviewer#2:

R#2: *It is rather puzzling that the visibility of the interference oscillations in the fractional quantum Hall regime (up to 60%) is larger than in the integer quantum Hall regime (about 40%). In my opinion, the authors have to convincingly explain this result, because if left unexplained, it would raise doubts about the general validity of their analysis. Moreover, the authors do discuss the low visibility in the integer quantum Hall regime and explain it with a mixing between co-propagating channels.*

This could be easily checked by measuring the visibility at filling factor 1, where only a single edge channel is present. These additional measurements would provide a more robust ground for the proposed explanation.

Besides, it is not explained why this process should be less relevant in the fractional QH regime, where for filling factor 2/5 there are again two co-propagating and interacting channels. In addition, it is stated, correctly, that Coulomb interactions for filling factor 2 cannot explain a reduced visibility, which is then ascribed to some interaction with impurities and spin-orbit interactions. However, some further discussion needs to be added: the GaAs/GaAlAs 2DEG material does not have a strong spin orbit component (in contrast to InAs or InSb); the mentioned theory reference deals with a single impurity and two

channels, while here one of the two channels is fully reflected; and in addition, I would expect an impurity distribution.

In the introduction and the conclusions, the authors underline that their results will provide guidance to understand previous results on interferometry in Mach-Zehnder (MZ) geometry in the fractional QH regime. However, this argument is rather weak, as in one case, two-quasiparticle interference is detected, while in the other, single-quasiparticle interference. Thus, the argument should be elaborated better. In addition, it is written that in MZ interferometers no oscillations in the fractional quantum Hall regime have been reported so far. This is somehow in contrast with the findings of Deviatov et al. (doi: 10.1209/0295-5075/100/67009) that were performed in the fractional regime: could the authors comment on this point?

Answer 2.2a: We thank Reviewer#2 for pointing out the non-intuitive experimental fact that the visibility at $\nu=2$ is found lower than that at $\nu=2/5$. We have identified the loss of visibility as resulting from channel mixing. As stated in the original main text this is a highly probable source of visibility loss, particularly as, contrary to what happens in MZIs, Coulomb interaction between co-propagating channels is not expected to affect the visibility, see M. Acciai et al. Phys. Rev. B 105, 125415 (2022). At $\nu=2$ channel mixing arises from disorder combined with the weak, but finite, spin-orbit interaction expected in GaAs (an effect much weaker than in InAs or InSb as correctly pointed by Reviewer#2). The loss of visibility is therefore not universal but sample dependent. As stated in the main text “typical scattering lengths ranging from few tens of μm to $100\ \mu\text{m}$ have been reported at filling factor 2 [26-28](now [29-31])” in 2DEGs realized in GaAs. We found a typical 10% mixing probability (see the supplementary material) while the distance between the injecting ohmic contacts and the QPC is about $20\ \mu\text{m}$. This suggests a few $100\ \mu\text{m}$ scattering length. This is the same order of magnitude of what has been reported in the literature and makes this scattering mechanism highly probable. Note that interchannel mixing should also be considered as a possible source of visibility loss in MZIs (see the added B3 part in the new supplementary information).

Reviewer#2 raises the question of considering a single mixing impurity rather than a continuous distribution of impurities. As a response to this comment, in the supplementary information, we have added a comparison between a single mixing point and a continuous distribution of weak mixing scatterers. We show that the two approaches give comparable results, see figure S7, showing that the global qualitative effect is visibility loss and it is not dependent on the choice of single or continuously distributed impurities as long as they are weak.

The fact that one channel is fully reflected, instead of fully transmitted, does not change the conclusion either: what matters is that one extra channel (reflected or transmitted) interacts with the interfering channels. The incoming quasiparticles have their wavefunction partly projected on the extra edge mode and the which-path information acquired by the extra channel induces a loss of interference visibility.

A comparison between $\nu=2$ (IQHE) and $\nu=2/5$ (FQHE) is delicate. The Jain picture of composite fermions provides an intuitive correspondence between filling factors 2 and $2/5$, but there is unfortunately no similar correspondence for the microscopic source of channel mixing. In a field of 11 Tesla, the $\nu=2/5$ FQHE ground state is fully spin polarized while, at a field of 2.3 Tesla, the $\nu=2$ IQHE ground state is not spin polarized so the channel mixing strength cannot be compared. A theory describing channel mixing originating from tunneling between two co-propagating fractional edge channels is missing. One can nevertheless expect that, from what we know from chiral Luttinger liquid physics, the low energy interchannel tunneling, being in the so-called strong backscattering regime, may be strongly inhibited. After re-analyzing the data, we do

not see any flaw in the measurements and we have to accept the higher visibility in FQHE as a non-universal but sample-dependent experimental fact.

As a response to this concern, we have included a discussion on the IQHE/FQHE visibility in the text.

Answer 2.2b: Reviewer#2 suggests to check the visibility for filling factor $\nu=1$. Indeed, at first sight, at integer filling only one channel carries the current. However, we think that considering filling factor 1 is of no help and brings more complexity. Indeed the smooth edge confinement leads to edge reconstruction where the $\nu=1$ channel breaks into several fractional edge channels. The same happens in a QPC. The common experience observed by several groups is that, because of the smooth saddle potential created by the split gates a spatially smooth density reduction occurs. This leads to a reconstruction of the $\nu=1$ channel preventing defining a single transmission D (the QPC conductance in units of e^2/h) as several conductance plateaus occur signaling locally formed $2/3, 1/3, \dots$, fractional states. For example, the standard Lesovik-Martin-Landauer-Büttiker shot noise expression proportional to $D(1-D)$ has never been found able to fit the noise data at $\nu=1$, while it correctly does at $\nu=2, 3$ and higher filling factors. A published study showing the complexity of QPC conductance and shot noise at bulk filling factor $\nu=1$ can be found, for example, in Phys. Rev. Lett. 114, 056802 (2015). Note that for this reason, quantum optics with electrons experiments in integer quantum Hall devices are standardly carried out at $\nu>1$.

Answer 2.2c: Reviewer#2 suggests to give a quantitative correspondence between the quality of interference measured in our two-particle interferometer and that observed in a single-particle MZI interferometer. A tentative answer is given above in the response to Reviewer#1's questions, who expressed a similar concern. In addition, we would like to point out that we did not claim that our experiment provides a "guidance" to understand previous results on MZIs, but rather that it indicates that propagating quasiparticle states in the FQHE do retain quantum coherence which is needed to observe interference, in contrast with many experiments with MZIs. To clarify further our comment about the role of mixing for single- and two-particle interferometry, we have added a section to the supplementary, discussing mixing in the MZI.

We also thank Reviewer#2 for providing the reference to the published work of E.V. Deviatov's team who reported MZ interference at $\nu=1/3$. We have included the reference and included discussion in the main text.

R#2: *More in detail:*

I suppose there is some mix-up in Figure 1(f). Contact (2) should be biased but is indicated as grounded.

Answer 2.3: thanks for noting this, the figure is now corrected.

R#2: *Page 4, line 20, the unit of mobility is m^2/Vs .*

Answer 2.3 : thanks, this is corrected.

R#2: *On page 4, the authors state that "We choose to only partition the inner edge channel while the outer edge channel is fully transmitted. Similar observations can be obtained when fully reflecting the inner edge and partitioning the outer edge." These results should be shown in the SI.*

Answer 2.4 : We have rephrased this sentence, as we realized that from the referee's comment the word "can" was not accurate and confusing. What we wanted to say is that "similar observations can *potentially* be obtained when permuting reflected and transmitted channels". These measurements have not been done as we thought that this rather trivial result would not have brought new information, while concentrating on the fractional regime would be of much higher value. We have adjusted the mentioned sentence in the revised manuscript.

R#2: *On page 6, when discussing typical scattering length for elastic tunneling between co-propagating edge channels, it would help to give the device size for comparison (which is only provided in the SI).*

Answer 2.5: This information has been added in the main text.

R#2: *At the beginning of the part on the FQHE regime, I wonder how useful it is to discuss the $\nu=1/3$ example when the experiments were performed at $\nu=2/5$ in the bulk. In addition, it is true that using a composite fermion picture, $\nu=2/5$ can be mapped onto $\nu=2$ filling with dressed entities. However, I think that this might be too trivial for the $\nu=2/5$ physics. For instance, the non-Fermi liquid behavior expected for fractional edge states results in a peculiar, and well-known, power-law behavior that is not mentioned at all. I think that the theory side should be thus revised, to avoid a too simple picture.*

Answer 2.6: In the text we have specialized the discussion on a Laughlin state at $\nu=1/3$ for pedagogical reason (and also we think that depletion in the QPC does realize a local $\nu=1/3$ Laughlin state). However, we understand from the Reviewer's remark that the choice of a Laughlin state may be inadequate as we are starting from a Jain's state, namely $\nu=2/5$. We have reformulated the sentence and we now focus on fractionally charged excitations. At bulk filling factor $\nu=2/5$ it is indeed possible to probe different charges in a single experiment. One can probe the $e/5$ edge excitations of the $2/5$ FQHE ground state. Thanks to the local reconstruction in the QPC, giving rise to a stable $1/3$ FQHE ground state, it is also possible to probe the local $e/3$ edge excitations of this ground state. This is why we think that it is useful to consider the $2/5$ experimental case.

In the weak backscattering regime, which is relevant for probing the quasiparticles, our experience is that we observe only weakly non-linear I-V characteristics. The theoretical fixed-point exponents predicted by LL theory are rarely observable. Indeed almost all experiments to date are done away from the fixed-point limit for practical reasons (a discussion of this can be found in <https://doi.org/10.1016/B978-0-12-822083-2.00003-4>, see figure 6.29 of this reference).

Because of the very weak non-linearity observed, we fit the DC shot noise using a constant transmission (this is mentioned in the legend of figure 3) or we fit the data using a non-interacting shot noise formula including a voltage dependent transmission (figure 4). The fits agree with the data within 10% accuracy. For the two-particle interference shot noise, we just have to use an accurate representation of the DC shot noise data (see Equation (6) of original main text).

Reviewer #3 (Remarks to the Author):

R#3: *This paper describes a Hanbury Brown and Twiss (HBT) experiment for fractional quasiparticles in the fractional quantum Hall regime. The cross-correlation noise under*

microwave excitations shows sinusoidal dependence on the time delay between the two microwave sources. The visibility is high (about 40% for integer charges and 60% for fractional charges), and this degradation is explained by “mixing” (tunneling) between the copropagating channels. The two-particle interferometry for fractional charges is demonstrated with a high-quality data set. This provides a novel characteristic of fractional charges. I have several questions before recommending this paper for publication in Nature Communications.

Answer 3: We thank Reviewer#3 for his/her positive comments on our work. Below we answer to his/her more detailed comments.

R#3: While the demonstrated scheme should ‘probe the HBT phase’ as shown in the text, this is not discussed with the experimental data. The authors should describe theoretically predicted values and experimentally demonstrated values of the HBT phase.

Answer 3.1: As explained in the original work of Büttiker and collaborators, the noise interference is a periodic function of $\phi - \chi$, where ϕ is the relative phase of the ac sources and χ the HBT phase. The actual value of χ is not universal as this relies on the sample geometry (propagation lengths) and edge mode velocity. Indeed $\chi = \arg(s_{13}^* s_{32} s_{24}^* s_{41})$ is $\pi + 2\pi f(l_{13} + l_{24} - l_{32} - l_{41})/v$ where l_{ij} is the distance between contacts i and j . According to the low added value of the concrete value of χ and the highly technical difficulty to know accurately the precise value of the electromagnetic phase difference between contacts of a sample placed in cryogenic high field environment, we decided not to perform a detailed characterization of this quantity. We have included a sentence in the main text to address this issue. A discussion about the HBT phase can be found in the review by M. Büttiker and P. Samuelsson "Interference of independently emitted electrons in quantum shot noise", *Ann. Phys.* 16 751 (2007).

R#3: For the fractional case, the ac amplitude of 650 and 870 μV in Fig. 3(c) is much larger than the voltage range (up to 250 μV) used for the DC shot noise in Fig. 3(b). The authors should confirm that fractional charge tunneling is allowed even for the large bias. The maximum bias can be $2 \cdot \sqrt{2}$ times the rms amplitude.

Answer 3.2: We thank Reviewer#3 for raising this point that we should have discussed in the original manuscript. In order to use equation (2) for IQHE or (6) for FQHE, we need to know the DC shot noise for voltages up to $V_{max} = l_{max} h f / e$ where l_{max} is the typical value of l for which the integer Bessel functions J_l^2 start to vanish. Typically, for a given $\Delta\phi$, $V_{max} = 2V_{ac} \sin(\frac{\Delta\phi}{2})$. While the DC shot noise has not been measured up to 650 or 870 μV , we have extrapolated the DC shot noise curve to higher values. This procedure was legitimate based on our experience on photo-assisted shot noise measurements in the FQHE regime for which we applied similar voltages, see M. Kapfer et al, *Science* (2019). In this work the DC shot noise remained linear for higher DC voltage (more than 500 μV) and for similar conditions. The good fit of the data confirmed this choice of extrapolating the DC shot noise curve. Note also that what matters is not voltage but energy: $e \cdot V$. For $V=870 \mu\text{V}$ the energy is $(1/m)870 \mu\text{eV}$ for $m=3$ or 5 . It is below the expected FQHE gaps for a high field of 11 Tesla. We have added a sentence in the text about the question of extrapolating the DC shot noise curve.

R#3: The supplementary data in Fig. S2 shows that the dip position shifts with the P2 power. Why is this?

Answer 3.3: The noise measurements were done over a very long time (it takes more than one day to obtain even few interference curves at different power). During this time, the relative electromagnetic phase $\Delta\phi$ generated by our room temperature microwave set-up drifts with time due to daily variation of temperature. We think it is this effect, and not the change of RF power, which is responsible for the slight phase shift in the data.

R#3: *Two notations, ΔF_{ai} and Δf_{ai} , are used for the relative phase.*

Answer 3.4: Thank you for pointing this out, we have fixed this point.

REVIEWERS' COMMENTS

Reviewer #2 (Remarks to the Author):

In my opinion, the authors have properly replied to all questions and comments raised by the referees, and thus I would now recommend publication of the manuscript in Nature Communications. There are a few minor issues, however, that the authors might want to consider before publication:

Page 2: "ascribed to the nature of states involved in the two interferometers: discrete versus continuous spectrum, the latter being more fragile with respect to environmental fluctuations." Why is the latter more susceptible to noise? Is it possible to add at least a reference?

Page 2: "The experiment is inspired from the seminal work of V. Rychkoff, Polianski, and M. Büttiker [12]". It should be either "V. S. Rychkoff, M. L. Polianski, and M. Büttiker" or "Rychkoff, Polianski, and Büttiker".

Page 6: "leads to a close expression" close -> closed

Page 8: There is an "(a)" missing in the caption of Figure 3: "Figure 3. (a) QPC conductance ..."

I am still puzzled by the low visibility in the integer quantum Hall regime, but the authors now properly discuss this issue on page 9. As a final suggestion for future work, if their interpretation is correct, a sample with a density such that $\nu=2$ is observed at 11 Tesla should show a much higher visibility.

Reviewer #3 (Remarks to the Author):

This paper describes a Hanbury Brown and Twiss (HBT) experiment for fractional quasiparticles in the fractional quantum Hall regime. The cross-correlation noise under microwave excitations shows sinusoidal dependence on the time delay between the two microwave sources. The visibility is high (about 40% for integer charges and 60% for fractional charges), and this degradation is explained by "mixing" (tunneling) between the copropagating channels. The two-particle interferometry for fractional charges is demonstrated with a high-quality data set. This provides a novel characteristic of fractional charges. I think the paper is revised satisfyingly to resolve the reviewers' criticisms. I would like to recommend this paper for publication in Nature Communications.

ANSWER TO REVIEWER COMMENTS

Reviewer#2 (Remarks to the Author):

In my opinion, the authors have properly replied to all questions and comments raised by the referees, and thus I would now recommend publication of the manuscript in Nature Communications. There are a few minor issues, however, that the authors might want to consider before publication:

Answer to Reviewer#2

We warmly thank Reviewer#2 for recommending publications. We have implemented his last suggestions and remarks which will improve the final version of the manuscript.

Page 2: "ascribed to the nature of states involved in the two interferometers: discrete versus continuous spectrum, the latter being more fragile with respect to environmental fluctuations." Why is the latter more susceptible to noise? Is it possible to add at least a reference?

Answer: we have added reference [16]: "Quantum coherence engineering in the integer quantum Hall regime", Phys. Rev. Lett. 108, 256802 (2012). Which demonstrates the role of providing a gap in the spectrum.

Page 2: "The experiment is inspired from the seminal work of V. Rychkoff, Polianski, and M. Büttiker [12]". It should be either "V. S. Rychkoff, M. L. Polianski, and M. Büttiker" or "Rychkoff, Polianski, and Büttiker".

Answer: done

Page 6: "leads to a close expression" close -> closed

Answer: done

Page 8: There is an "(a)" missing in the caption of Figure 3: "Figure 3. (a) QPC conductance ..."

Answer: done

I am still puzzled by the low visibility in the integer quantum Hall regime, but the authors now properly discuss this issue on page 9. As a final suggestion for future work, if their interpretation is correct, a sample with a density such that $\nu=2$ is observed at 11 Tesla should show a much higher visibility.

Answer: thanks for the suggestion. We are planning realizing higher density sample to check the visibility in this different regime.

Reviewer #3 (Remarks to the Author):

This paper describes a Hanbury Brown and Twiss (HBT) experiment for fractional quasiparticles in the fractional quantum Hall regime. The cross-correlation noise under microwave excitations shows sinusoidal dependence on the time delay between the two microwave sources. The visibility is high (about 40% for integer charges and 60% for fractional charges), and this degradation is explained by "mixing" (tunneling) between the copropagating channels. The two-particle interferometry for fractional charges is demonstrated with a high-quality data set. This provides a novel characteristic of fractional charges. I think the paper is revised satisfyingly to resolve the reviewers' criticisms. I would like to recommend this paper for publication in Nature Communications.

Answer to Reviewer#3: We warmly thank Reviewer#3 for recommending publication and for his/her time to review our work. The remarks and criticisms were very useful in improving the clarity of the presentation.